# Racial and ethnic disparities in workers' compensation claims rates

**Caroline K. Smith**[ID]*, **Sara Wuellner, Jennifer Marcum**

Washington State Department of Labor & Industries, Safety & Health Assessment & Research for Prevention (SHARP) Research Program, Olympia, Washington, United States of America

* Caroline.Smith@LNI.wa.gov

**Data Availability Statement:** Data that support the findings of this study are not publicly available due to privacy and ethical restrictions. Data contain personal identifying and sensitive information, and per Washington State Law RCW 51.28.070, claim files and records are confidential. A limited de-

## Abstract

### Background

Workers of color experience a disproportionate share of work-related injuries and illnesses (WRII), however, most workers' compensation systems do not collect race and ethnicity information, making it difficult to monitor trends over time, or to investigate specific policies and procedures that maintain or could eliminate the unequal burden of WRII for workers of color. The purpose of this study is to apply a Bayesian method to Washington workers' compensation claims data to identify racial and ethnic disparities of WRII by industry and occupation, improving upon existing surveillance limitations. Measuring differences in risk for WRII will better inform prevention efforts and target prevention to those at increased risk.

### Methods

To estimate WRII by race/ethnicity, we applied the Bayesian Improved Surname Geocode (BISG) method to surname and residential address data among all Washington workers' compensation claims filed for injuries in 2013–2017. We then compare worker and injury characteristics by imputed race/ethnicity, and estimate rates of WRII by imputed race/ethnicity within industry and occupation.

### Results

Black/African Americans had the highest rates of WRII claims across all industry and occupational sectors. Hispanic/Latino WRII claimants also had higher rates than Whites and Asian/Pacific Islanders in almost all industry and occupational sectors. For accepted claims with both medical and non-medical compensation, Bodily reaction/overexertion injuries accounted for almost half of the claims during this reporting period.

### Discussion

The high rates of injury we report by racial/ethnic categories is a cause for major concern. Nearly all industry and occupation-specific rates of workers' compensation claims are higher for Black/African American and Hispanic/Latino workers compared to Whites. More work is needed to identify work-related, systemic, and individual characteristics.

identified dataset is attached in supplemental files. Dr. David Bonauto is our organizations contact for access to identifiable data (David.Bonauto@LNI. wa.gov).

**Funding:** The authors received no specific funding for this work.

**Competing interests:** The authors have declared that no competing interests exist.

## Introduction

Rates of non-fatal WRII have been reported to vary by race and ethnicity, although estimates based on large, representative datasets are rare. Instead, evidence that workers of color bear a disproportionate share of WRII often come from two types of studies. The first are narrow in scope by industry, injury outcome, geography, or time period [1–3], limiting their ability to monitor trends over time or to make comparisons across worker groups. In the second type, evidence of disparities is based on group-level comparisons rather than case-level data. Injury data by industry and occupation are compared to employment data by race/ethnicity, leading researchers to conclude that non-white workers are at higher risk of workplace injuries because they are more likely to be employed in industries and occupations with high rates of WRII. In this second approach, differences in WRII by race/ethnicity within industries cannot be measured [4, 5]. A better understanding of existing disparities in WRII by race and ethnicity within industries and occupations is crucial to developing and targeting injury and illness prevention efforts and advancing health equity [6, 7].

Most large-scale data sources routinely used for monitoring nonfatal WRII are insufficient for estimating risks by worker race/ethnicity [8]. The Bureau of Labor Statistics annual Survey of Occupational Injuries and Illnesses (SOII) collect information on race/ethnicity for cases involving days of missed work, but the race/ethnicity data are missing for over 40% of cases [9]. The Behavioral Risk Factor Surveillance System (BRFSS) can be used for state estimates of WRII by race/ethnicity when states include an optional work injury module in the survey, but single year sample sizes are not sufficient for producing estimates by race/ethnicity in combination with other factors including industry, so multiple years must be combined, which means that timely surveillance of rates of injury by race/ethnicity can't be accomplished with BRFSS [10]. The nationally representative population-based National Health Interview Study (NHIS) utilizes a large sample size, and has been used to assess differences in WRII by race/ethnicity [11, 12], but as with BRFSS, estimates of WRII are dependent on a work supplement included sporadically. Medical care data such as hospital discharge or emergency department data may capture race/ethnicity–often as perceived by the health care provider rather than as reported by the patient–but these sources generally lack information on the injured worker's industry or occupation.

When self-reported race/ethnicity is not available, the Bayesian Improved Surname Geocode method (BISG) developed by the RAND Corporation, can be used to estimate race/ethnicity based on last name and residential address data [13, 14]. Using the racial and ethnic population distributions within census block groups in combination with proportions by last names listed in the Census Surname File, BISG is used to calculate a probability of belonging to the following six mutually exclusive categories: (1) White, (2) Black/African American, (3) American Indian/ Alaska Native, (4) Asian/Pacific Islander, (5) "More than one race," and (6) Hispanic/Latino (all races).

Workers' compensation claims data is an administrative dataset that usually lacks self-reported race/ethnicity, but does include worker name and address data–the information needed to estimate race/ethnicity using BISG. Washington State's workers' compensation data include additional information detailing the injury event and severity, the industry of the injured worker's employer, and the injured worker's demographic characteristics. The purpose of this study is to apply BISG to Washington workers' compensation claims data to identify racial and ethnic disparities of WRII by industry and occupation, improving upon the surveillance limitations identified earlier. Measuring differences in risk for WRII will better inform prevention efforts and target prevention to those at increased risk.

## Materials and methods

Data used in this study come from the 1) The Washington State Department of Labor & Industries, 2) The American Community Survey, and 3) the Census Surname List, and 4) the Current Population Survey (CPS), via the Centers for Disease Control Employed Labor Force (ELF) online query, all of which are described below.

### Washington's workers' compensation system

All Washington State employers are required to obtain workers' compensation insurance unless workers are covered by an alternative workers' compensation system (e.g., the federal government, employers of railroad, or long-shore workers), or are specifically exempted in Washington statute, such as the self-employed [15]. Approximately 182,000 employers and 3 million workers are insured through the Washington State Department of Labor and Industries (WA L&I) industrial insurance system, referred to as the State Fund [16]. The remaining covered Washington State workers (25 percent) are employed by approximately 350 employers that self-insure for workers' compensation [16]. The Department administers all State Fund claims, and receives regulatory data on all workers' compensation claims for self-insured employers.

Claims are classified as either rejected, the claim did not meet the requirements for the worker to receive medical or time-loss compensation; accepted for medical aid only, the worker received benefits for medical costs only; or accepted for both medical and non-medical costs–including time-loss compensation, permanent disability awards, survivors' benefits, funeral expenses, and/or pension benefits. Accepted claims are assigned Occupational Injury and Illness Coding System (OIICS) v.1 codes [17] based on information on the Report of Industrial Injury or Occupational Disease/Report of Accident (ROA), a form filed by the health care provider and injured or ill worker to initiate a workers' compensation claim. These claims are assigned an OIICS code for the nature of the injury, source of injury, body part affected, and type of event (or exposure) that led to the injury. We aggregated results by the 1-digit OIICS event code [17].

### Workers' compensation claim data collection

Washington State Fund and self-insured workers' compensation claims were included in this study. The data were abstracted from the system on July 23, 2019, and included claims with an injury date from 2013 through 2017 (n = 1,004,068) to ensure alignment with the ACS residential address data used. We extracted claimant surname and residential address for the BISG, and additional key administrative variables including the following: age, gender (binary male/female), OIICS injury event (injury type), claimant language preference, National Occupational Research Agenda (NORA) [18] industry sector, Standard Occupation Classification (SOC) major group, and claim adjudication status (rejected or accepted).

### Census data

The American Community Survey is a product of the U.S. Census Bureau, and replaces the long form decennial census. The ACS collects detailed information about population and housing characteristics as well as economic, social and demographic data. For this study, the racial ethnic distributions by Census Block Group were downloaded from the American Community Survey (ACS) 2017 5-year summary file for Washington State (2013–2017) [19]. We obtained race/ethnicity proportions for the six mutually exclusive categories above by surname from the 2010 Decennial Census Surname list [20] (Census, 2016). Surnames must have been

reported, along with self-reported race/ethnicity, at least 100 times in the 2010 Census to be included in the Census Surname list.

Denominator data for rate calculations were obtained from the Current Population Survey (CPS), using the Centers for Disease Control and Prevention (CDC), National Institutes for Occupational Safety and Health (NIOSH), Employed Labor Force (ELF), query system [21] to estimate full time equivalent workers (FTE) within each industry sector during 2013 through 2017.

### Bayesian Improved Surname Geocoding (BISG) method

This study used the Bayesian Improved Surname Geocode (BISG) method developed by Elliott et al. [14] and detailed elsewhere [22] to estimate race/ethnicity among Washington workers' compensation claimants. Briefly, the BISG method uses racial/ethnic proportions by geographical area of residence, and surname from the U.S. Bureau of the Census (Census) to calculate posterior probabilities of belonging to six mutually exclusive racial/ethnic categories. Each record is assigned the posterior probabilities of being (1) White, (2) Black/African American, (3) American Indian/ Alaska Native, (4) Asian/Pacific Islander, (5) "More than one race," and (6) Hispanic. The posterior probabilities for each record are then summed across the six categories to estimate the race/ethnicity distribution in the sample. The BISG method is not designed to assign race/ethnicity to individual records [14]. While the BISG estimates probabilities for American Indian/Alaska Native and "Multiracial," prior studies demonstrate that the BISG produces less robust estimates for these racial groups [14, 22, 23], therefore results for these two racial/ethnic groups will not be reported.

The general equation for calculating Bayes' theorem is calculated as the probability of A given B (prior), which is the probability of B given A, multiplied by the probability of A. This is divided by the probability of B or written as:

$$P(A|B) = \frac{P(B|A)P(A)}{P(B)}$$

For each claimant, we estimate the probability of being Black given the proportion of residents in their block group who identify as Black, multiplied by the proportion of Americans with their surname who identify as Black. This is done for each additional race/ethnicity category: (1) White, (2) Black/African American, (3) American Indian/ Alaska Native, (4) Asian/ Pacific Islander, (5) "More than one race," and (6) Hispanic.

### Analysis

Workers' compensation claimant surnames were matched to the Census Surname List to obtain proportions of race/ethnicity by surname. Claims with surnames that did not match the Census Surname List were excluded from analyses (n = 92,268, 9.2%). The Washington Master Addressing Services (WAMAS) in the Washington State Office of the Chief Information Officer, standardized addresses to the USPS standard format and geocoded the claimant addresses using ArcGIS for this project [24]. Claimants with no residential address information, who did not live in Washington State, or had a P.O. Box or rural road address were excluded from analyses (n = 92,466, 9.2%). Elliott and colleagues do not recommend using P.O. Boxes or rural road addresses for the BISG method because it reduces accuracy of the estimation [14]. All remaining claimants were then assigned a 12-digit Federal Information Processing System (FIPS) code for Census Block Group. Claimants were matched to the ACS dataset with proportions of race/ethnicity groups by Census Block Group.

Using Bayes' Theorem, we then calculated the posterior probabilities of a claimant being the following four race/ethnicity categories (1) White, (2) Black/African American, (3) Asian/

Pacific Islander, and (4) Hispanic/Latino for each claim remaining in the sample (n = 819,234). The posterior probabilities were then summed for the race/ethnicity categories by the administrative variables listed above to compare characteristics of filed claims by imputed race/ethnicity (e.g., age, gender, NORA industry sector, SOC occupation major group, etc.).

Workers' compensation claim rates were then calculated by race/ethnicity NORA industry sector, and SOC major groups (the 22 SOC major groups were combined in our analyses due to small numbers in some groups). Numerators were calculated by multiplying the summed posterior probabilities by the total number of claims in each industry sector. Denominator data for rate calculations were obtained from the CPS, using the Centers for Disease Control and Prevention (CDC), National Institutes for Occupational Safety and Health (NIOSH), Employed Labor Force (ELF), query system [21] to estimate full time equivalent workers (FTE) within each occupation and industry sector during 2013 through 2017. Claim incidence rates are presented per 1,000 FTE, and 95% Confidence Intervals are presented with claim rates. The confidence intervals were calculated using the delta method [25] (Klein 1953) to derive variance.

The Washington State Department of Labor & Industries is a public health authority, and as such is allowed by 45 CFR 46.102(/)(2) to conduct public health surveillance to identify, monitor, assess or investigate conditions of public health importance. The work presented here, falls within public health surveillance activities as covered by the Common Rule, and is not required to be approved by an institutional review board; therefore, no institutional review board approval was sought. A minimal dataset is available in the Supporting Information files. Access to additional confidential data can be requested by contacting David Bonauto (David. Bonauto@LNI.wa.gov) at the Washington State Department of Labor and Industries Safety & Health Assessment & Research for Prevention (SHARP) Program. Access to additional accompanying confidential data would need to adhere to Washington State restrictions to protect patient confidentiality (Revised Code of Washington 51.28.070) and subject to review by the Washington State Institutional Review Board (see Revised Code of Washington 42.48 and 49.17.210).

## Results

Table 1 describes the claims studied, by estimated race/ethnicity. Hispanic/Latinos had the lowest percentage of rejected claims, while Asian/Pacific Islanders and Black/African Americans had the highest. Hispanic/Latinos had the highest proportion of claims accepted for medical aid only, and the lowest proportion of accepted for medical and non-medical compensation. There were only minor differences by gender and age. Hispanic/Latinos were substantially less likely to prefer communication in English (55% versus 81–99%). The most common injury types were bodily reaction and overexertion injuries, accounting for about 1/2 of all claims accepted for medical and non-medical compensation injury types during this period. Hispanic/Latino's had the lowest proportion of injuries classified as bodily reaction and overexertion (41.8%), and Whites had the highest (51.4%).

Table 2 presents rates of injury among accepted claims by race/ethnicity within each NORA industry sector, as well as rate ratios comparing all other racial/ethnic groups to Whites. Rates for all non-White groups were significantly different than rates among Whites for all industry sectors. Black/African Americans had considerably higher rates of accepted claims compared to Whites in every NORA sector ranging from a fourteen-fold higher rate in Agriculture, Forestry, and Fishing to a 2.3 to 8.4 fold higher rate in all other sectors. Rate ratios for Hispanic/Latino accepted claims compared to Whites were also higher in every NORA

**Table 1. Characteristics of all filed claims in Washington State 2013–2017 by estimated race/ethnicity.**

| | Black/African American | Hispanic/ Latino | Asian/Pacific Islander | White |
|---|---|---|---|---|
| | N = 70369 | N = 165404 | N = 32874 | N = 530693 |
| | % (95% CI) | % (95% CI) | % (95% CI) | % (95% CI) |
| Claim Status | | | | |
| Rejected | 16.9 (16.2, 17.6) | 12.3 (11.9, 12.8) | 17.6 (16.7, 18.6) | 14.9 (14.6, 15.1) |
| Accepted for medical aid only | 52.0 (51.5, 52.5) | 60.7 (60.4, 61.0) | 52.8 (52.1, 53.5) | 55 (54.8, 55.2) |
| Accepted for medical and non-medical compensation | 31.1 (30.4, 31.7) | 27.0 (26.6, 27.4) | 29.5 (28.6, 30.5) | 30.1 (29.9, 30.4) |
| Gender | | | | |
| Female | 39.6 (39.0, 40.2) | 32.4 (32.0, 32.8) | 42.7 (41.9, 43.5) | 36.3 (36.0, 36.5) |
| Male | 60.4 (59.9, 60.8) | 67.6 (67.3, 67.8) | 57.3 (56.6, 58.0) | 63.7 (63.6, 63.9) |
| Age Groups | | | | |
| 16–24 years old | 13.4 (12.7, 14.1) | 16.1 (15.6, 16.5) | 11.5 (10.5, 12.5) | 13.8 (13.6, 14.1) |
| 25–34 years old | 25.6 (24.9, 26.2) | 28.3 (27.9, 28.7) | 23.3 (22.4, 24.3) | 22.8 (22.6, 23.0) |
| 35–44 years old | 20.9 (20.2, 21.5) | 25.7 (25.2, 26.1) | 21.5 (20.5, 22.4) | 19.7 (19.5, 20.0) |
| 45–54 years old | 22.1 (21.4, 22.8) | 18.3 (17.9, 18.7) | 22.3 (21.4, 23.3) | 22.6 (22.4, 22.8) |
| 55–64 years old | 14.3 (13.7, 15.0) | 8.8 (8.3, 9.2) | 16.7 (15.7, 17.7) | 17.0 (16.7, 17.2) |
| 65+ years old' | 2.4 (1.6, 3.1) | 1.6 (1.1, 2.0) | 3.1 (2.1, 4.2) | 2.9 (2.6, 3.1) |
| Language preference[b] | | | | |
| English[c] | 97.8 (97.7, 97.9) | 55.1 (54.8, 55.4) | 81.2 (80.7, 81.6) | 98.9 (98.8, 98.9) |
| Spanish | 0.9 (0.1, 1.6) | 44.6 (44.3, 45.0) | 4.5 (3.5, 5.6) | 0.8 (0.6, 1.1) |
| Other | 1.1 (0.4, 1.9) | 0.2 (0.0, 0.7) | 2.9 (1.9, 4.0) | 0.1 (0.0, 0.4) |
| Vietnamese | 0.0 (0.0, 0.8) | 0.0 (0.0, 0.5) | 5.4 (4.3, 6.4) | 0.0 (0.0, 0.3) |
| Chinese | 0.0 (0.0, 0.8) | 0.0 (0.0, 0.5) | 3.0 (2.0, 4.1) | 0.0 (0.0, 0.3) |
| Korean | 0.0 (0.0, 0.8) | 0.0 (0.0, 0.5) | 1.5 (0.4, 2.6) | 0.0 (0.0, 0.3) |
| Russian | 0.0 (0.0, 0.8) | 0.0 (0.0, 0.5) | 0.0 (0.0, 1.1) | 0.1 (0.0, 0.4) |
| Cambodian | 0.0 (0.0, 0.8) | 0.0 (0.0, 0.5) | 1.2 (0.1, 2.2) | 0.0 (0.0, 0.3) |
| Laotian | 0.0 (0.0, 0.7) | 0.0 (0.0, 0.5) | 0.2 (0.0, 1.3) | 0.0 (0.0, 0.3) |
| Injury type[d] | | | | |
| Bodily Reaction and Overexertion | 50.5 (49.6, 51.4) | 41.8 (41.1, 42.5) | 48.2 (46.8, 49.6) | 51.4 (51.1, 51.7) |
| Contact with Objects and Equipment | 15.6 (14.4, 16.8) | 21.4 (20.6, 22.3) | 18.5 (16.7, 20.3) | 15.1 (14.6, 15.5) |
| Falls | 17.8 (16.6, 19.0) | 25.0 (24.2, 25.8) | 17.5 (15.7, 19.3) | 18.5 (18.1, 19.0) |
| Exposure to harmful substances or environments | 2.7 (1.4, 4.0) | 2.4 (1.5, 3.3) | 2.9 (0.9, 4.8) | 3.6 (3.1, 4.1) |
| Assaults and Violent acts | 3.3 (2.0, 4.6) | 1.7 (0.8, 2.6) | 2.4 (0.4, 4.4) | 2.8 (2.3, 3.2) |
| Transportation incidents | 4.3 (3.0, 5.6) | 2.6 (1.7, 3.5) | 3.2 (1.2, 5.1) | 3.1 (2.6, 3.6) |
| Other events or exposures | 5.8 (4.5, 7.1) | 5.0 (4.1, 5.9) | 7.3 (5.4, 9.2) | 5.5 (5.0, 6.0) |
| Fire and explosions | 0.0 (0.0, 1.4) | 0.1 (0.0, 1.0) | 0.0 (0.0, 2.0) | 0.1 (0.0, 0.5) |
| NORA Sector Group[e] | | | | |
| Agriculture, Forestry, & Fishing | 0.7 (0.0, 1.4) | 23.6 (23.2, 24.0) | 0.9 (0.0, 2.0) | 1.9 (1.6, 2.1) |
| Construction | 8.9 (8.2, 9.6) | 12.5 (12.1, 13.0) | 5.6 (4.6, 6.7) | 11.5 (11.2, 11.8) |
| Healthcare & Social Assistance | 15.4 (14.7, 16.0) | 8.1 (7.6, 8.6) | 16.4 (15.4, 17.4) | 13.2 (12.9, 13.5) |
| Manufacturing | 10.3 (9.6, 11.0) | 11.2 (10.7, 11.6) | 15.4 (14.4, 16.4) | 11.6 (11.3, 11.8) |
| Mining, Gas & Oil Extraction | 0.1 (0.0, 0.8) | 0.1 (0.0, 0.6) | 0.0 (0.0, 1.1) | 0.2 (0.0,0.5) |
| Services | 39.7 (39.1, 40.3) | 27.0 (26.6, 27.4) | 38.4 (37.6, 39.3) | 38.0 (37.8, 38.2) |
| Transportation, Warehousing, and Utilities | 7.1 (6.4, 7.8) | 3.5 (3.0, 3.9) | 5.1 (4.1, 6.2) | 5.5 (5.2, 5.7) |
| Wholesale/Retail Trade | 16.5 (15.9, 17.2) | 13.1 (12.7, 13.6) | 16.7 (15.7, 17.7) | 16.9 (16.7, 17.2) |
| Not classifiable/missing | 1.4 (0.7, 2.1) | 0.9 (0.4, 1.4) | 1.5 (0.4, 2.5) | 1.3 (1.0, 1.5) |
| Occupational Groups[f] | | | | |
| Construction, Extraction, Maintenance and Repair | 15.1 (14.4, 15.8) | 17.0 (16.6, 17.5) | 10.6 (9.6, 11.6) | 16.9 (19.4, 19.8) |

*(Continued)*

**Table 1.** (Continued)

| | Black/African American | Hispanic/ Latino | Asian/Pacific Islander | White |
|---|---|---|---|---|
| | N = 70369 | N = 165404 | N = 32874 | N = 530693 |
| | % (95% CI) | % (95% CI) | % (95% CI) | % (95% CI) |
| Farming, Fishing, and Forestry | 0.5 (0.0, 1.2) | 17.7 (17.2, 18.1) | 0.7 (0.0, 1.8) | 1.3 (1.0, 1.5) |
| Management and Professional | 11.1 (10.4, 11.8) | 4.5 (4.0, 5.0) | 12.6 (11.6, 13.6) | 12.4 (12.2, 12.7) |
| Production, Transportation, and Material Moving | 24.7 (24.1, 25.4) | 23.5 (23.1, 24.0) | 24.0 (23.1, 25.0) | 22.0 (21.8, 22.2) |
| Sales and Office Occupations | 10.1 (9.4, 10.8) | 4.9 (4.4, 5.4) | 9.0 (8.0, 10.1) | 9.7 (9.4, 10.0) |
| Service Occupations | 24.6 (24.0, 25.3) | 21.5 (21.1, 22.0) | 27.0 (26.1, 27.9) | 22.3 (22.1, 22.5) |
| Not classifiable/missing | 13.8 (12.8, 14.8) | 10.9 (10.0, 11.7) | 16.0 (14.6, 17.5) | 12.7 (12.3, 13.0) |

[a] Rejected claims include claims not approved for compensation whether due to incorrect or missing information, or denial of work-relatedness. Accepted for medical aid only are those claims that qualified for compensation of medical care only, typically caused by less-severe injuries or illnesses. Claims accepted for medical and non-medical compensation are typically more severe and received reimbursement for time off work or disability.

[b] Language preference is from the administrative record, if an injured worker would prefer a language for written and verbal communication with the Department in a language other than English.

[c] English language preference is assumed if no non-English languages are indicated in the administrative data.

[d] Injury type is coded from the initial report of injury/illness using the one-digit Occupational Injury or Illness Classification System (OIICS) system for injury event (injury type). In addition, includes only accepted claims with medical and non-medical costs.

[e] Services sector includes Public Safety.

[f] Bureau of Census Occupations 2-digit, cross walked to Standard Occupation Classification (SOC) 2-digit major groups.

Sector ranging from 1.2 for compensable claims in Construction, to a rate ratio of 2.2 in Agriculture, Forestry and Fishing. Asian/Pacific Islanders had similar or lower rates compared to Whites in all NORA sectors, except for Agriculture, Forestry, and Fishing (rate ratio 3.9).

Table 3 provides rates by two-digit 2010 Standard Occupational Classification (cross walked from 2010 Bureau of Census (BOC) codes). Table 3 rate ratios are similar to the industry level rate ratios presented in Table 2, with Black/African Americans having the highest rates and rate ratios (compared to Whites) across all occupational groups, followed by Hispanic/Latino workers. Overall, Asian/Pacific Islanders had lower rates by occupation than Whites.

**Table 2. Rates of accepted workers' compensation claims by estimated race/ethnicity and NORA industry sectors[a].**

| NORA Sectors | Black/African American | | Hispanic/Latino | | Asian/Pacific Islander | | White | |
|---|---|---|---|---|---|---|---|---|
| | Rate per 1,000 FTE (95% CI) | Rate ratio[b] | Rate per 1,000 FTE (95% CI) | Rate ratio[b] | Rate per 1,000 FTE (95% CI) | Rate ratio[b] | Rate per 1,000 FTE (95% CI) | Rate ratio[b] |
| Agriculture, Forestry & Fishing | 924.0 (923.2, 924.8) | 14.0 | 145.5 (142.6, 148.4) | 2.2 | 257.5 (257.2, 257.9) | 3.9 | 65.8 (64.8, 66.7) | 1.0 |
| Construction | 706.7 (705.3, 708.1) | 8.4 | 103.8 (103.8, 103.9) | 1.2 | 82.8 (82.5, 83.1) | 1.0 | 84.5 (82.9, 86.0) | 1.0 |
| Healthcare | 114.2 (113.5, 114.9) | 2.3 | 68.3 (67.7, 68.9) | 1.4 | 23.1 (22.9, 23.3) | 0.5 | 49.5 (48.3, 50.7) | 1.0 |
| Manufacturing | 109.5 (108.8, 110.1) | 2.7 | 82.5 (81.6, 83.4) | 2.0 | 21.0 (20.8, 21.2) | 0.5 | 41.2 (40.1, 42.3) | 1.0 |
| Mining, Gas & Oil Extraction | - | - | -[c] | - | - | - | 30.4 (21.4, 39.5) | 1.0 |
| Services | 115.9 (115.5, 116.3) | 2.9 | 66.9 (66.5, 67.3) | 1.7 | 16.2 (16.1, 16.3) | 0.4 | 39.4 (38.8, 40.1) | 1.0 |
| Transportation, Warehouse & Utilities | 133.5 (132.2, 134.7) | 2.9 | 68.8 (67.8, 69.8) | 1.5 | 27.4 (27.1, 27.7) | 0.6 | 46.7 (44.9, 48.5) | 1.0 |
| Wholesale and Retail Trade | 157.8 (157.1, 158.4) | 3.1 | 94.5 (93.8, 95.2) | 1.8 | 21.6 (21.4, 21.7) | 0.4 | 51.1 (50.1, 52.2) | 1.0 |

[a] To correspond with CDC Employed Labor Force data, Mining was combined with Gas & Oil, and Services includes Public Safety.

[b] Rate ratio displays rates for non-White groups compared to White.

[c] "–" indicates insufficient numbers to calculate rates.

**Table 3. Rates of accepted workers' compensation claims by estimated race/ethnicity and Standard Occupational Classification (SOC) major groups.**

| SOC major groups | Black/African American | | Hispanic/Latino | | Asian/Pacific Islander | | White | |
|---|---|---|---|---|---|---|---|---|
| | Rate per 1,000 FTE (95% CI) | Rate ratio[a] | Rate per 1,000 FTE (95% CI) | Rate ratio[a] | Rate per 1,000 FTE (95% CI) | Rate ratio[a] | Rate per 1,000 FTE (95% CI) | |
| Rate ratio[a] | | | | | | | | |
| Construction, Extraction, Maintenance, and Repair | 567.1 (566.0, 568.1) | 5.4 | 114.5 (113.8, 115.2) | 1.1 | 87.7 (87.5, 88.0) | 0.7 | 104.6 (103.2, 105.9) | 1.0 |
| Farming, Fishing, and Forestry | 1224.5 (1223.4, 1225.5) | 15.0 | 128.3 (125.1, 131.6) | 1.6 | 126.4 (126.1. 126.6) | 1.6 | 81.5 (80.3, 82.7) | 1.0 |
| Management and Professional | 45.0 (44.6, 45.5) | 3.3 | 21.2 (20.9, 21.5) | 1.6 | 4.7 (4.6, 4.8) | 0.3 | 13.5 (12.7, 14.2) | 1.0 |
| Production, Transportation, and Material Moving | 224.6 (223.9, 225.3) | 2.3 | 129.5 (128.8, 130.30 | 1.3 | 47.7 (47.4, 47.9) | 0.5 | 96.8 (95.7, 98.0) | 1.0 |
| Sales and Office | 67.9 (67.2, 68.5) | 3.1 | 30.7 (30.2, 31.2) | 1.4 | 12.7 (12.5, 12.8) | 0.6 | 22.1 (21.1, 23.1) | 1.0 |
| Service | 154.7 (154.1, 155.2) | 1.4 | 96.5 (95.9, 97.1) | 0.9 | 40.9 (40.7, 41.1) | 0.4 | 111.9 (110.6, 113.1) | 1.0 |

[a]Rate ratio displays rates for non-White groups compared to White.

Comparing claims included in the analysis (those with estimated race/ethnicity) and claims excluded for not having either surname from the Census surname list, or and address that was not block group codeable, there were differences (at least a 5% difference between study data and data missing) for claim, and injury type category. Differences were also found by claim type, with fewer claims accepted for medical aid only missed due to residential address not being codeable, compared to missing surnames. However, larger differences by claim type, language preference, NORA Sector, and SOC occupations were found, results are presented in Table 4. Among the differences, we found a lower percent of Spanish language claimants' last names were not on the Census Surname list, (4.5% compared to those that were BISG coded 9.9%). Almost 12% of workers in the Agriculture, Forestry, and Fishing NORA sector were not included in the data set due to missing residential address data, and 18% in the Services SOC occupation group were not included, again due to not-codeable address.

## Discussion

To our knowledge, this is the first population-based study to report workers' compensation claim rates by race/ethnicity within industry sector and occupation groups. Previous studies of occupational injury disparities found that a greater proportion of workers of color are employed in high hazard industries compared with white workers, suggesting that workers of color are at greater risk of workplace injury because of employment patterns [3–5]. To arrive at their conclusions, researchers combined occupational injury data by industry from one source with race/ethnicity data by industry from another source, linked at the level of industry and not the individual worker. This approach requires one to assume that injury risk is equal across all workers within an industry or occupation. Using BISG, we found that risk of WRII was not equal across workers in the same industry sector or occupational group. Within each industry sector, rates of WRII varied by race/ethnicity, with Black/African American workers experiencing the highest rates, and Asian/Pacific Islanders generally experiencing the lowest rates. Additionally, the magnitude of the difference in rates across race/ethnicity differed by industry and occupation, with the greatest differences in rates by race/ethnicity observed among the highest risk industries and occupations, notably Agriculture, Forestry, and Fishing. The ultimate goal for occupational safety and health professionals is to reduce all work-related injuries to zero, and while we have come a long way since the inception of OSHA, this goal is still far from a reality. That Black/African American and Hispanic/Latino workers have

**Table 4. Differences in characteristics between the coded BISG sample, and those not coded due to missing or non-specific physical address and those with less common surnames.**

| | BISG coded (N = 819218) | Unable to geocode address (N = 102479) | Unable to code surname (N = 92254) | Significant[e] |
|---|---|---|---|---|
| | N (%) | N (%) | N (%) | |
| Claim type[a] | | | | |
| Rejected | 120044 (14.7) | 19600 (19.1) | 15180 (16.5) | |
| Accepted for medical aid only | 456932 (55.8) | **47276(46.1)** | 49750 (53.9) | *** |
| Accepted for medical and non-medical compensation | 242152 (29.6) | **35603 (34.7)** | 27324 (29.6) | *** |
| Gender | | | | |
| Female | 295329 (36.1) | 35947 (35.0) | 36054 (39.1) | |
| Male | 523811 (63.9) | 66618 (65.0) | 56204 (60.9) | |
| Age groups | | | | |
| 16–24 years old | 115898 (14.2) | 12453 (12.3) | 13220 (14.3) | |
| 25–34 years old | 198323 (24.2) | 21678 (21.4) | 22200 (24.1) | |
| 35–44 years old | 173102 (21.1) | 21366 (21.1) | 18760 (20.3) | |
| 45–54 years old | 177490 (21.7) | 24559 (24.3) | 19932 (21.6) | |
| 55–64 years old | 123182 (15.0) | 17909 (17.7) | 14115 (15.3) | |
| 65+ years and older | 20935 (2.6) | 3143 (3.1) | 2525 (2.7) | |
| Language preference[b] | | | | |
| English[c] | 730506 (89.2) | 90022 (87.8) | 84607 (91.7) | |
| Spanish | 80868 (9.9) | 11940 (11.6) | **4190 (4.5)** | *** |
| Other | 2897 (0.4) | 315 (0.31) | 1724 (1.9) | |
| Vietnamese | 1912 (0.2) | 85 (0.1) | 41 (0.0) | |
| Chinese | 1111 (0.1) | 54 (0.1) | 34 (0.0) | |
| Korean | 729 (0.1) | 29 (0.0) | 23 (0.0) | |
| Russian | 560 (0.1) | 112 (0.1) | 1444 (1.6) | |
| Cambodian | 533 (0.1) | 17 (0.0) | 65 (0.1) | |
| Laotian | 92 (0.0) | 10 (0.0) | 140 (0.2) | |
| Injury type[d] | | | | |
| Bodily Reaction and Overexertion | 279304 (34.1) | **17175 (48.2)** | 13698 (50.1) | *** |
| Contact with Objects and Equipment | 230453 (28.1) | **5387 (15.1)** | 4525 (16.6) | *** |
| Falls | 121829 (14.9) | **7104 (20.0)** | 4941 (18.1) | *** |
| Exposure to harmful substances or environments | 45469 (5.6) | 1244 (3.5) | 831 (3.0) | |
| Assaults and Violent acts | 29073 (3.6) | 847 (2.4) | 781 (2.9) | |
| Transportation incidents | 22363 (2.7) | 1109 (3.1) | 836 (3.1) | |
| Other events or exposures | 90266 (11.0) | 2721 (7.6) | 1695 (6.2) | |
| Fire and explosions | 451 (0.1) | 16 (0.0) | 17 (0.1) | |
| NORA Sector Group | | | | |
| Agriculture, Forestry, & Fishing | 50070 (6.1) | **11748 (11.5)** | 2960 (3.2) | *** |
| Construction | 91939 (11.2) | 9916 (9.7) | 8664 (9.4) | |
| Healthcare & Social Assistance | 110701 (13.5) | 12736 (12.4) | 15163 (16.4) | |
| Manufacturing | 94460 (11.5) | 11716 (11.4) | 10338 (11.2) | |
| Mining, Gas & Oil Extraction | 1253 (0.2) | 218 (0.2) | 76 (0.1) | |
| Services | 287443 (35.1) | 31871 (31.1) | 33634 (36.5) | |
| Transportation, Warehousing, and Utilities | 42567 (5.2) | 6190 (6.0) | 5482 (5.9) | |
| Wholesale/Retail Trade | 132018 (16.1) | 15185 (14.8) | 14628 (15.9) | |
| Not classifiable/missing | 8757 (1.1) | 3004 (2.9) | 1323 (1.4) | |

*(Continued)*

**Table 4.** (Continued)

| | BISG coded (N = 819218) | Unable to geocode address (N = 102479) | Unable to code surname (N = 92254) | Significant[e] |
|---|---|---|---|---|
| | N (%) | N (%) | N (%) | |
| SOC Major Groups | | | | |
| Construction, Extraction, Maintenance, and Repair | 149857 (18.3) | 16661 (16.2) | 14927 (16.2) | |
| Farming, Fishing, and Forestry | 36799 (4.5) | 9255 (9.0) | 2157 (2.3) | |
| Management and Professional | 87592 (10.7) | 10412 (10.2) | 11161 (12.1) | |
| Production, Transportation, and Material Moving | 185659 (22.7) | 21896 (21.3) | 20351 (22.1) | |
| Sales and Office | 71627 (8.7) | 7556 (7.4) | 8392 (9.1) | |
| Service Occupations | 184839 (22.6) | **18022 (17.6)** | 22117 (24.0) | *** |
| Not classifiable/missing | 102835 (12.6) | **18782 (18.3)** | 13163 (14.3) | *** |

[a] Rejected claims include claims not approved for compensation whether due to incorrect or missing information, or denial of work-relatedness. Accepted for medical aid only are those claims that qualified for compensation of medical care only, typically caused by less-severe injuries or illnesses. Claims accepted for medical and non-medical compensation are typically more severe and received reimbursement for time off work or disability.

[b] Language preference identified by health care provider or injured worker.

[c] English is indicated when no language is entered into the administrative database.

[d] Injury types for accepted for medical and non-medical compensation

[e] Inidcates a difference ≥ 5 percentage points from the coded BISG sample.

historically been over-represented [3, 26] in work-related fatal and non-fatal injuries regardless of industry or occupation must be more than a passing description of results, it must become significant focus for all public health professionals aiming for health equity.

Racism in housing, education, hiring and occupational mobility have all been shown to reduce the life chances of Black and Hispanic/Latino populations in America [4, 27, 28], and these issues combine to determine which jobs and occupations a person has access to. Limited by educational opportunities, residential segregation, discrimination in hiring and lack of mobility, workers of color are more likely sorted into high hazard jobs [27, 29, 30], and this study found that even *within* occupation and industry, rates of injury differ significantly by race and ethnicity. These multiple forms of harm to historically marginalized racial and ethnic groups in the United States largely determine our individual and societal health, income, wealth and future generational realities, it is imperative we disrupt these overlapping vulnerabilities if (workplace) health equity is our goal.

While this study adds significantly to the literature, by addressing industry sector, occupation, and race/ethnicity disparities in injury rates, it does have some notable limitations. First, the racial/ethnic probabilities were indirectly derived by using a Bayesian statistical method, not self-reported. While the BISG has been shown to be robust method for identifying racial/ethnic group membership [14, 22], it is likely not as precise as self-report. In addition, due to limitations of the BISG, we were not able to report American Indian/Alaska Natives, or More than One Race injury rates. Third, the BISG provides simultaneous probabilities for very general races and Hispanic/Latino ethnicity, but cannot distinguish between different sub groups such as Chinese or Cambodian, Peruvian or Puerto Rican, or distinct sovereign nation affiliations, nor can it distinguish between native and foreign-born workers. Finally, the missing address and surname data that resulted in many workers' compensation claims being dropped from analysis, appears to have reduced the number of Spanish speakers in this study, which most likely underestimates the rates and rate ratios for Hispanics/Latinos. Future research

should address these limitations and potentially add more information such as first name [31] and other data such as language spoken, which could improve the BISG estimates for the broad racial and Hispanic/Latino groups.

Claim filing behavior is also another important limitation, Fan et al. [32] used data from the Washington State Behavioral Risk Factor Surveillance System, and found that, with the exception of American Indians, non-White workers were less likely to file a workers' compensation claim, although the difference was not found to be statistically significant; differences between self-reported injuries and filing a workers' compensation claim varied considerably by occupation, with farming/forestry/fishing ranked highest in reporting a work-related injury or illness, but second lowest in filing a workers' compensation claim. If workers of color are less likely than White workers to file a workers' compensation claim, disparities in WRII by race and ethnicity are likely greater than estimated here.

Finally, it is possible that the differences in rates of WRII by race/ethnicity may still be due to segregation into high hazard industries and occupations, obscured by aggregation into the broad categories of industry and occupation used in this study. Estimates of injury rates for more detailed industry groupings would require population estimates of workers by race/ethnicity and industry based on sample sizes larger than the current CPS or some other robust source of employment data by industry and race/ethnicity (ACS involves a larger sample but produces estimates of employed persons and not FTE).

These limitations are noteworthy, but considering the lack of alternatives in the field of occupational health to quantify work injury disparities by race/ethnicity, industry, and occupation, this study still contributes a great deal to the ongoing discussion of racial/ethnic disparities in occupational safety and health. The high rates of injury we report by racial/ethnic categories is a cause for major concern. The working population in Washington state during this data period was 73% White, 3.3% Black/African American, 1.2% American Indian/Alaska Native, 10.2% Asian/Pacific Islander, 12.2% Hispanic/Latino, while the rates per 1,000 FTE are higher for all non-White racial and Hispanic/Latino ethnicity, compared to Whites. More work is needed to identify work-related, systemic, and individual characteristics. We need to improve our understanding of what creates these inequities to add to the discussion of reducing the disparities of work-related injuries and illnesses. Adding self-reported race and ethnicity to the workers' compensation system would be an ideal way estimate racial and ethnic inequities, however even if we were to implement this today, it would take years to have enough self-reported data to address some of the major findings in this study. Adding self-reported race and ethnicity to workers' compensation systems would in the long run, allow us to identify more nuanced ethnic inequities that could lead to better targeting of interventions.

## Conclusion

To our knowledge, no workers' compensation system in the United States collects data on the race and ethnicity of claimants, which makes it difficult to identify disparities in injury risk, health service access, utilization, and outcomes by these critical social constructs. Enumerating inequities in workers' compensation claim rates is foundational to identifying and addressing root causes of disparities in WRII. Utilizing a well-tested method for indirect estimation of race and ethnicity, we found that risk of WRII is not similar within an industry or occupation, and that certain industries and occupations have larger racial/ethnic disparities in who is at risk for injury. Understanding injury risk by race/ethnicity can better allocate resources for prevention, elicit new lines of research and provide researchers and policymakers with much needed knowledge of how racism might be affecting workplace safety and workers' compensation insurance programs. It is the first step toward identifying policies, procedures and laws

that need to be dismantled, re-imagined, or created so that all workers, regardless of race and ethnicity, come home safe and healthy from work each day.

## Supporting information

**S1 Dataset.**
(7Z)

## Author Contributions

**Conceptualization:** Caroline K. Smith, Sara Wuellner, Jennifer Marcum.

**Data curation:** Caroline K. Smith.

**Formal analysis:** Caroline K. Smith.

**Methodology:** Caroline K. Smith, Sara Wuellner, Jennifer Marcum.

**Writing – original draft:** Caroline K. Smith, Sara Wuellner, Jennifer Marcum.

**Writing – review & editing:** Caroline K. Smith, Sara Wuellner, Jennifer Marcum.

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
