## [Decision Letter · Decision Letter 0]

26 Sep 2022

PONE-D-22-19151Racial and ethnic disparities in workers' compensation claims ratesPLOS ONE

Dear Dr. Smith,

Thank you for submitting your manuscript to PLOS ONE. After careful consideration, we feel that it has merit but does not fully meet PLOS ONE’s publication criteria as it currently stands. Therefore, we invite you to submit a revised version of the manuscript that addresses the points raised during the review process.

The format of this manuscript and the format of the reference style should be identical with the format of PLOS ONE. Please check the guideline of PLOS ONE. There lacks conclusion section, please add this section. The contribution of the study and the research gap should be identified and illustrated. All tables should have frames, please check the standard format of PLOS ONE. The literature review section is too short and should be extended. Please ensure that your decision is justified on PLOS ONE’s publication criteria and not, for example, on novelty or perceived impact.

We look forward to receiving your revised manuscript.

Kind regards,

Jing Cheng

Academic Editor

PLOS ONE

Journal Requirements:

2. You indicated that ethical approval was not necessary for your study. We understand that the framework for ethical oversight requirements for studies of this type may differ depending on the setting and we would appreciate some further clarification regarding your research. Could you please provide further details on why your study is exempt from the need for approval and confirmation from your institutional review board or research ethics committee (e.g., in the form of a letter or email correspondence) that ethics review was not necessary for this study? Please include a copy of the correspondence as an ""Other"" file.

4. Please ensure that you include a title page within your main document. As per our author guidelines (http://journals.plos.org/plosone/s/submission-guidelines#loc-title-page) we do require this to be part of the manuscript file itself.

5. Please amend your manuscript to include your abstract after the title page.

Reviewers' comments:

Reviewer's Responses to Questions

**Comments to the Author**

1. Is the manuscript technically sound, and do the data support the conclusions?

Reviewer #1: Partly

Reviewer #2: Yes

Reviewer #3: Yes

2. Has the statistical analysis been performed appropriately and rigorously? 

Reviewer #1: No

Reviewer #2: Yes

Reviewer #3: Yes

3. Have the authors made all data underlying the findings in their manuscript fully available?

Reviewer #1: No

Reviewer #2: No

Reviewer #3: No

4. Is the manuscript presented in an intelligible fashion and written in standard English?

Reviewer #1: Yes

Reviewer #2: Yes

Reviewer #3: Yes

5. Review Comments to the Author

Reviewer #1: The manuscript (MS) aims to apply a Bayesian Improved Surname Geocode (BISG) Method to identify racial/ethnic disparities of workers’ compensation claims by industry and occupation in Washington State. To address this aim, the authors used claims from 2013 through 2017 (n=1,004,068) to ensure alignment with the American Community Survey (ACS) residential address data. While the MS focuses on an important and understudied theme, there are some content and methodological issues compromising the inferences made out its results.

Specific comments:

1) Abstract: Aim could be clearly stated; Methods section should include the sample or population used for the analyses; the Results could benefit from some estimates and should specifically address the aim; the Discussion section is long and could be more succinct with regards to the policy implications of the findings.

2) Introduction: This section is long and a bit disorganized. As is, every paragraph seems disconnected. There is not a clear or strong rationale for why workers’ compensation matter for racial/ethnic health inequities and what the potential mechanisms may be for the racial/ethnic inequities. The aim is better stated in this section.

3) Methods: This section provides some details on the methods but not enough details on the data used and the variables considered (Table 1 presents a number of variables that were never mentioned in this section). How different were the records excluded from those included? Can they be compared? This seems to be in table 4, please include the actually N in the column headings.

4) Analyses: The section needs to be presented in a way that anyone having access to the data could replicate the analyses. Moreover, the presentation should reflect the results presented in the tables.

5) Results: The tables present a lot of data. However, the interpretation was thin. Please make sure the titles of the tables included information on who, where and when for the data. It is unclear what the difference is between tables 2 and 3.

6) Discussion: This section includes several statement lacking references or out of context. For instance, the paragraph starting in line 208 alludes to previous studies but it does not include references. In fact, the entire paragraph does not include a single reference. The paragraph starting in line 227 goes on about racism but provides not context on why this is connected to the findings. Please discuss the implications of the limitations for the findings, i.e., under or overestimation. What is the advantage of using BISG after all? The last paragraph of the Discussion does not seem a good one to close the paper.

Overall, the Discussion section has very few references and seems more supported by speculation than facts and evidence.

Reviewer #2: I enjoyed reading this article that uses a novel approach to describe racial and ethnic disparities among Washington workers’ compensation claims. This is an area where there is a dearth of data, so efforts to address this are much needed. While this methodology may not be perfect, you did a good job acknowledging the limitations and providing data analysis to further describe potential biases. Overall, I think this was very nicely written and had only a few minor comments.

1. Paragraph lines 52-64: Since there are mutually exclusive, specify whether race or ethnicity takes precedence. For example, if you are Black/African American and Hispanic, which one are you coded as?

2. Line 101: Extra comma needs removed after “rejected.”

3. Line 228: Change “Blacks” to “Black”

Of note, while the statistical methods applied appear to be fine to me, I am not a statistical expert and not qualified to make a formal assessment of the statistical methods applied.

Reviewer #3: This is an excellent and seminal research study that identified racial and ethnic disparities in workers' compensation claims rates. The below suggestions may further enhance the work.

•Although the RAND method has been validated, were there any attempts by the authors to validate a small random sample of algorithm-coded WA WC claims through a call-back survey, linkage to another data set, or by other means? While not required, this may help to bolster future work.

•For the next steps and discussion, could the authors comment about the possibility and pros/cons of adding race and ethnicity data into the actual future WC claims collection process in Washington? This has been a topic with renewed interest, and having one state take the lead in this process could influence other states to collect similar data.

6. PLOS authors have the option to publish the peer review history of their article (what does this mean?). If published, this will include your full peer review and any attached files.

Reviewer #1: No

Reviewer #2: No

Reviewer #3: No

---

## [Author Response · Author response to Decision Letter 0]

13 Dec 2022

Reviewer Comment number Reviewer’s comment Authors’ response

1 1 Abstract: Aim could be clearly stated; Methods section should include the sample or population used for the analyses; the Results could benefit from some estimates and should specifically address the aim; the Discussion section is long and could be more succinct with regards to the policy implications of the findings.

 Thank you. We have added detail regarding the aim of the paper, and included the population used for analyses.

We are not sure that your comments here for the results and discussion section of the abstract but in response: We are under the belief that numbers and statistical data should not be included in the abstract. The discussion is three sentences long, we have edited the discussion by eliminating the last sentence. 

1 2 Introduction: This section is long and a bit disorganized. As is, every paragraph seems disconnected. There is not a clear or strong rationale for why workers’ compensation matter for racial/ethnic health inequities and what the potential mechanisms may be for the racial/ethnic inequities. The aim is better stated in this section.

 Thank you, we strive to be as parsimonious as possible. We believe that readers of this manuscript will not have as much knowledge of worker related injury and illness data limitations, therefore we feel strongly that these limitations need to be stated in the introduction, as it points to the rationale for the paper.

We have edited the introduction in an attempt to make it more focused. The following is the order of our paragraphs and arguments within.

Paragraph 1:

Work related injuries and illness are more likely in high hazard industries, and occupations.

We need better data by race and ethnicity for work-related injuries

Paragraph 2:

Most data for racial/ethnic rates of work related injury are lacking –in scope or scale or both.

Paragraph 3:

Describes how large national datasets are lacking as they are incomplete (lack sufficient data, no denominator, etc.)

Paragraph 4:

How we can overcome the limitations of data described in paragraphs 2 and 3. 

Paragraph 5:

Introduces the data we will use to estimate racial/ethnic rates of injury, and why it is a better dataset for estimating rates of work related injury/illness than national datasets.

1 3 Methods: This section provides some details on the methods but not enough details on the data used and the variables considered (Table 1 presents a number of variables that were never mentioned in this section). How different were the records excluded from those included? Can they be compared? This seems to be in table 4, please include the actually N in the column headings.

 Thank you, we have re-organized the Methods section (now called Materials and Methods), and describe more fully all of the data sources used in the analyses.

All variables in table 1 are included in the Methods section: Claim status, Injury event (injury type), age, gender, industry and occupation. We have added “(injury type)” to the existing definition of “injury event” in order to clarify what that variable is.

We describe, in detail the Washington state workers’ compensation data – who is covered, how the data is entered into the system, the types of claims, when data was abstracted, the number of claims and the years of those claims. We are not sure what additional information could be added. In addition we identified the data used in addition to the workers’ compensation data – that is the American Community survey data, and the Census surname list. These are both publically available. We have added hyperlinks in the references to make it easier for readers to get that information. 

You are correct, the comparisons between the data included and excluded are in table 4 and discussed in the results.

Total N for each column in Table 4 has been added.

1 4 Analyses: The section needs to be presented in a way that anyone having access to the data could replicate the analyses. Moreover, the presentation should reflect the results presented in the tables.

 Thank you, we have added the Bayesian calculation to the Analysis section, so that others can duplicate our results. 

We have re-ordered the presentation to reflect the results presented in the tables.

1 5 Results: The tables present a lot of data. However, the interpretation was thin. Please make sure the titles of the tables included information on who, where and when for the data. It is unclear what the difference is between tables 2 and 3.

 Multiple occupations exist in each industry, and national data frequently describes industry level exposure, risk and rates of injury, however this sometimes obscures specific occupations that may be at risk, regardless of the industry they work in. Therefore, we present the information separately for Industry (table 2) and Occupation (table 3). Both tables make this distinction in the titles. The manner in which industries and occupations are labeled in the National coding systems, do sound similar at the Major group level. 

1 6 Discussion: This section includes several statement lacking references or out of context. For instance, the paragraph starting in line 208 alludes to previous studies but it does not include references. In fact, the entire paragraph does not include a single reference. The paragraph starting in line 227 goes on about racism but provides not context on why this is connected to the findings. Please discuss the implications of the limitations for the findings, i.e., under or overestimation. What is the advantage of using BISG after all? The last paragraph of the Discussion does not seem a good one to close the paper. Overall, the Discussion section has very few references and seems more supported by speculation than facts and evidence. Great catch, thank you. We have added the references that should have been listed in this paragraph. 

We have added context to the paragraph you noted, to add clarity as to why we are making the claim that racism is connected to inequity rates of injury in this study. 

2 1 . Paragraph lines 52-64: Since there are mutually exclusive, specify whether race or ethnicity takes precedence. For example, if you are Black/African American and Hispanic, which one are you coded as? Excellent question and possibly one of the reasons Elliott et al (2009) do not recommend using this to identify a single person’s race or ethnicity. Each individual is given a posterior probability of belonging to each race and Latino ethnicity (the race/ethnicity categories are non-Hispanic white, non-Hispanic black, non-Hispanic API, Hispanic (all races)). We therefore have in this paper, 4 probabilities for each record, this way we do not have to choose/prioritize one over the other. To estimate rates – we sum all probabilities for each race and Latino ethnicity to calculate the “N” for each, so each person has all of the probabilities used.

The posterior probabilities calculate the probability of being Black/African American for each individual. They also calculate the probability of being Hispanic/Latino for each individual – meaning that each race/ethnicity alone – is given a mutually exclusive probability. So one individual has probabilities for being White, API, Black/African American, Hispanic/Latino, etc., so it is each individual probability that is race/ethnicity alone, not the individual person. 

2 2 Line 101: Extra comma needs removed after “rejected.” Thank you, this has been removed.

2 3 Line 228: Change “Blacks” to “Black” Thank you, this has been corrected.

3 1 Although the RAND method has been validated, were there any attempts by the authors to validate a small random sample of algorithm-coded WA WC claims through a call-back survey, linkage to another data set, or by other means? While not required, this may help to bolster future work. This is a good idea. We have validated this using self-reported data from previous studies (see Smith and Bonauto, 2018), and while not specifically WC population, they are workers within Washington State, so the overall population is similar. 

3 2 For the next steps and discussion, could the authors comment about the possibility and pros/cons of adding race and ethnicity data into the actual future WC claims collection process in Washington? This has been a topic with renewed interest, and having one state take the lead in this process could influence other states to collect similar data. Excellent suggestion. We have added this to our discussion of next steps.

---

## [Decision Letter · Decision Letter 1]

27 Dec 2022

Racial and ethnic disparities in workers' compensation claims rates

PONE-D-22-19151R1

Dear Dr. Smith,

We’re pleased to inform you that your manuscript has been judged scientifically suitable for publication and will be formally accepted for publication once it meets all outstanding technical requirements.

Kind regards,

Jing Cheng

Academic Editor

PLOS ONE

Additional Editor Comments (optional):

Reviewers' comments:

Reviewer's Responses to Questions

**Comments to the Author**

1. If the authors have adequately addressed your comments raised in a previous round of review and you feel that this manuscript is now acceptable for publication, you may indicate that here to bypass the “Comments to the Author” section, enter your conflict of interest statement in the “Confidential to Editor” section, and submit your "Accept" recommendation.

Reviewer #2: All comments have been addressed

Reviewer #3: All comments have been addressed

2. Is the manuscript technically sound, and do the data support the conclusions?

Reviewer #2: Yes

Reviewer #3: Yes

3. Has the statistical analysis been performed appropriately and rigorously? 

Reviewer #2: Yes

Reviewer #3: Yes

4. Have the authors made all data underlying the findings in their manuscript fully available?

Reviewer #2: No

Reviewer #3: Yes

5. Is the manuscript presented in an intelligible fashion and written in standard English?

Reviewer #2: Yes

Reviewer #3: Yes

6. Review Comments to the Author

Reviewer #2: Thank you for your detailed responses and corrections to all reviewer comments. From my perspective, all identified concerns have been addressed.

Reviewer #3: The authors have been responsive to all reviewers. The paper will highlight the need to research the issue of collection of race and ethnicity in the occupational systems moving forward.

7. PLOS authors have the option to publish the peer review history of their article (what does this mean?). If published, this will include your full peer review and any attached files.

Reviewer #2: No

Reviewer #3: No

---

## [Editor Report · Acceptance letter]

5 Jan 2023

PONE-D-22-19151R1 

Racial and ethnic disparities in workers’ compensation claims rates 

Dear Dr. Smith:

I'm pleased to inform you that your manuscript has been deemed suitable for publication in PLOS ONE. Congratulations! Your manuscript is now with our production department. 

Kind regards, 

on behalf of

Dr. Jing Cheng 

Academic Editor

PLOS ONE